# Deploying a Fotonovela to Combat Methamphetamine Abuse among South Africans with Varying Levels of Health Literacy

**DOI:** 10.3390/ijerph18126334

**Published:** 2021-06-11

**Authors:** Burt Davis, Carel J. M. Jansen

**Affiliations:** 1Africa Centre for HIV/AIDS Management, Faculty of Economic and Management Sciences, Stellenbosch University, Stellenbosch 7602, South Africa; 2Department of Communication and Information Sciences, Faculty of Arts, University of Groningen, 9712 EK Groningen, The Netherlands; C.J.M.Jansen@rug.nl; 3Language Centre, Stellenbosch University, Stellenbosch 7600, South Africa

**Keywords:** photo story, fotonovela, entertainment-education, methamphetamine, tik, substance abuse, health literacy, narratives, health information, health communication

## Abstract

Poor health literacy in the Western Cape Province of South Africa is one of the main factors hampering methamphetamine (MA) use prevention efforts in the area, where the abuse of this drug is a major health and social problem affecting especially previously disadvantaged communities. In the first part of a two-part study, we compared a health-related fotonovela about MA to an existing brochure group and a control group. Main findings show that the vast majority of readers preferred the fotonovela over the existing brochure. This included participants from all three age groups and for both levels of health literacy (low/high) distinguished (*n* = 372). Furthermore, specifically for older people with low levels of health literacy, the fotonovela outperformed the existing brochure condition for knowledge level. In the second part of the study, we found that healthcare providers (*n* = 75) strongly prefer a fotonovela over an existing brochure, while this cohort viewed the potential use of fotonovelas in a health care setting as very positive. Our findings add to the promising results of an earlier fotonovela study about MA use in South Africa, providing further support for considering using narratives in health communication as a serious option to effectively communicate convincing health information about this drug to target audiences in the Western Cape Province.

## 1. Introduction

### 1.1. The Problem of Methamphetamine Use

Methamphetamine (MA) use is associated with many detrimental physical and mental effects such as general bodily deterioration, extreme weight loss, anxiety, psychosis, and aggression [1,2]. At a community level, MA can also have several negative effects. Such is the case in the Western Cape Province of South Africa, where the alarming rise in MA use from 2002 to 2006/7 has since become a major social problem [3] (p. 33), [4,5]. With more users in the Western Cape Province than in any other South African province, MA—locally referred to as ‘tik’—is linked with higher rates of domestic violence and crime in this province, as well as with gang culture and the spread and contracting of tuberculosis [2,6,7,8,9]. Exacerbating the problem, is that the majority of people in South Africa who use drugs such as MA are already part of marginalized and vulnerable populations [3] (p.36).

Numerous substance abuse prevention campaigns and other programmes by entities such as the City of Cape Town municipality and the local Department of Social Development as well as various NGOs have been employed in the Western Cape Province over the last few years [10,11]. Despite these efforts, recent statistics show that MA use accounted for 44% of patient admissions at treatment centres/programs in this province in the first half of 2020—the highest of any drug. This was a significant increase compared to the previous reporting period (July–December 2019) when 29% of patient admissions at treatment centres/programs were due to MA use (also the highest of any drug during that period) [12]. Current health messaging interventions are seemingly not having the desired effect. Below, we firstly propose why this may be the case, and then explore the potential of an alternative health education tool to support current information dissemination efforts about MA use in the Western Cape Province.

### 1.2. Effects of Low Levels of Education and Literacy

A lack of evidence-based prevention strategies, irregular representative surveys on substance abuse, and the general under-reporting of drug use are some of the factors making it difficult to plan and develop targeted prevention interventions about MA use in the Western Cape Province [6,13]. Another likely factor hampering prevention efforts relates to the general lack of education and low literacy levels in the province.

In the Western Cape Province, in 2017, 2.18 million people had not completed high school (49.7% of the population), while 364,000 people 20 years or older from this cohort had not completed Grade 7 or had received no schooling at all (8.29% of the population) [14] (p. 79). Of these 364,000 people, 47,000 individuals could not write their own name, 83,000 people could not read, 136,000 people could not fill in a form, and 54,000 people could not calculate how much change they should receive [14] (p. 87). Many persons in the province thus can be considered as functional illiterate: they do not possess a minimum literacy level required for adequate participation in their respective society (for this and other definitions of functional illiteracy, see the systematic review by Vágvölgyi and colleagues [15]).

Closely related to functional literacy is the concept of health literacy [16]. Health literacy can be defined as: “the ability to perform basic reading tasks required to function in the health care environment, such as reading labels on prescription bottles, understanding information on appointment slips, completing health insurance forms and following instructions pertaining to diagnostic tests” [17] (p. 1677), or in short as: “the degree to which individuals have the capacity to obtain, process, and understand basic health information and services needed to make appropriate health decisions” [18]. Low health literacy has been reliably associated with negative health outcomes, such as poorer health knowledge, and poor overall health status [19,20,21]. Taken together, low levels of education, literacy, and associated health literacy have a negative effect on the ability of vulnerable and at-risk populations in the Western Cape to properly process, comprehend, and internalize basic health messages about MA use, which may partly explain why the prevalence of patient admissions related to this drug at treatment centres in the province is still on the rise at present.

### 1.3. Entertainment-Education to Counter the Effects of Low Health Literacy Levels

A strategy that has shown promise to deliver persuasive health-related prevention messages to audiences with low levels of health literacy, is entertainment-education (E-E). E-E embeds educational messages within popular entertainment content, with the aim of increasing knowledge, influencing attitudes and intentions, changing norms, or affecting behaviour change [22,23].

One type of printed E-E narratives is the fotonovela or photo storybook. Fotonovelas are pocket-size booklets with posed photographs and text bubbles/captions. Especially romantic fotonovelas have a tradition that goes back to the period just after World War II. The very first fotonovelas appeared in Italy, as cheap adaptations of popular films that people often could not afford to go see. In the next two decades, the phenomenon grew into a mass medium, especially in Spanish speaking countries. Since then, the fotonovela has virtually disappeared in Western Europe. In Central and South America and in the southern part of the United States, however, romantic photonovelas have remained in demand.

In 2009, Baron and his colleagues working at the University of Southern California came up with the idea of capitalizing on the popularity of fotonovelas among Hispanics in California, and using the fotonovela as a tool in health communication with specifically low-literacy groups. Their first fotonovela, titled *Sweet Temptations*, aimed to effectively inform readers about the dangers of diabetes, and also to entice them, if possible, to make behavioural choices that would be better for their own health and for that of people in their immediate environment. From pretest to posttest, a statistically significant increase was found in diabetes knowledge and intentions to exercise, eat fruits and vegetables, and talk to doctors and family members about diabetes [24].

Since then, a number of photonovelas on other health education topics have also been produced by Baron’s team. One example is *Secret Feelings*, on the theme of depression. The goal of this fotonovela was threefold: to increase readers’ knowledge of the symptoms and effects of depression, to direct them to agencies that can provide help, and to reduce the stigma with which the phenomenon of depression is still often surrounded. Compared to a low-literacy text pamphlet on depression, *Secret Feelings* was found to produce larger reductions in antidepressant stigma and mental health care stigma. The fotonovela also was more likely to be passed on to family or friends after the study [25,26,27].

In the last two decades, fotonovelas aimed at health education purposes have also been developed and tested in other countries. In South Africa, James et al. examined the effects of *Laduma*, a fotonovela about STDs, among high school students. Compared to the initial situation and also compared to a control group, there were clear gains in knowledge and also in attitudes toward STD prevention in the group that read *Laduma*. Students who had read *Laduma* said they were more likely to plan to use condoms in the next year [28]. In the Netherlands, Koops van ‘t Jagt et al. compared the effects of *Zoete verleiding*, a Dutch version of *Sweet Temptations*, to a traditional brochure on diabetes. In terms of knowledge about diabetes, participants in the fotonovela condition performed best. This finding was consistent across all literacy levels. However, on behavioural intentions, readers in the fotonovela did not score significantly higher than other participants [21].

Other health topics covered in fotonovelas include dementia [29], obesity, asthma, breast cancer, prescription drugs, and the human papillomavirus [30,31]. However, on the subject of drug use, as far as we know, no fotonovelas have been developed as yet, apart from the South African fotonovela discussed below [11,32].

### 1.4. An Earlier Study into the Possible Advantages of a Fotonovela to Combat Methamphetamine Abuse in South Africa

In a recent study by Davis and Jansen [11] in South Africa, a fotonovela was tested as a medium to communicate health-related prevention messages about MA. In this study, the fotonovela *Spyt kom te laat* (*Regret fixes nothing*) was developed, which was then compared to a no-message control group and a traditional brochure. In a randomized controlled trial (*n* = 303), the fotonovela outperformed the control condition for knowledge level and it outperformed the traditional brochure in terms of conversation prompting (intention to speak about MA). When asked if the participants preferred the fotonovela or the traditional brochure, a clear preference for the fotonovela was found. Especially readers with relatively low levels of education, as well as readers belonging to a younger age group (19 years or younger) and readers belonging to an older age group (35 and older) favoured the fotonovela.

Apparently, using an E-E approach in the form of fotonovelas could be a good fit to help boost the effectiveness of prevention efforts about MA use in the Western Cape Province on at least two levels. Firstly, the possible positive effects of fotonovelas for audiences with low levels of health literacy are well documented (see above). Using a fotonovela could therefore help to counterbalance the effects that inadequate health literacy has on these target audiences’ ability to read and understand the health messages presented.

Secondly, as a result of MA being the most common drug used by especially persons younger than 20 years in the Western Cape Province since 2004 [12], many current MA use prevention campaigns in this province not only target affected youths but also their parents. This is done in order to educate parents about dangers of MA which their children are often exposed to [33]. In a systematic review of empirical studies into health literacy of older adults, Zamora and Clingerman [34] conclude that advancing age results in a significant increase in prevalence of inadequate health literacy. As Koops van ‘t Jagt and colleagues [35] argue, for older adults, a low level of health literacy may even have more severe consequences than for younger adults.

The inherent capacity of fotonovelas to counter the effects of poor health literacy seems to be what is needed in the Western Cape Province. Fotonovelas can help current MA-related health messaging efforts to reach young people and their parents who are struggling to function in a health care environment. However, before fotonovelas can be considered as a serious option in current health communication strategies, more evidence is needed to properly compare their effectiveness with the effectiveness of other formats in conveying health information on MA.

### 1.5. Knowledge Gaps after the Initial Study

The results of the earlier study by Davis and Jansen [11] are promising, but not yet conclusive. To mention one important limitation, in this study, the literacy level of the participants was not measured. No clear, empirically based distinction was made between readers with low and high levels of health literacy. Furthermore, there were some psychometric limitations with regards to the items used to measure knowledge level, behavioural intentions, and behavioural attitudes, such as single-item measurements and possible ceiling effects. Moreover, the fotonovela in this study was not compared with an existing health brochure used in the field, but rather with a researcher-designed brochure. In addition, there was no differentiation between versions of the fotonovela with and without a Q&A section; only a fotonovela without a Q&A section was used. However, including such a Q&A section in a fotonovela may add to the positive effects of a fotonovela. As Koops van ‘t Jagt and colleagues [21] remarked, referring to Moyer-Gusé, Jain, and Chung [36], the combination of an entertainment narrative with an explicit persuasive appeal in an epilogue may positively contribute to influencing health behaviour. Finally, no data were collected on the views of healthcare providers in the Western Cape Province on the potential of fotonovelas as a means to educate people in their target groups.

Given these remaining knowledge gaps, we decided to gather more empirical data to reach an informed conclusion about the effects of a fotonovela intended to convincingly convey health information about MA.

### 1.6. Research Questions for the New Fotonovela Study about MA Use in South Africa

The first research question (RQ1) explored how a fotonovela (with or without a Q&A section) compared to an existing brochure in affecting MA knowledge, attitudes, intentions, and preference of readers from different age groups, with a low or a high level of health literacy. To answer RQ1, the target group consisted of general community members from areas where MA use is known to be a problem. This cohort included younger and older people from all walks of life, such as learners, out-of-school youths, and employed and unemployed adults.

We also wanted to find out how a MA fotonovela compared to an existing health brochure and how it would be evaluated when assessed by healthcare providers (RQ2). To answer RQ2, the target group comprised of professional nurses and community health workers who deal with the problem of MA use in affected communities. Community health workers, supported by other health professionals such as doctors and nurses, form a bridge between communities and healthcare service provision within health facilities in South Africa [37] (pp. 1–2).

### 1.7. Structure of the Rest of the Article

First, the materials and method for the part of the study related to RQ1 will be discussed. This will be followed by the materials and the method for the part of the study related to RQ2. Then, the results for RQ1 will be presented, followed by the results for RQ2. The article concludes with a general discussion.

## 2. Methods 

### 2.1. Method for the Part of the Study Related to RQ1

A four-armed randomized controlled trial (RCT) was conducted, including four message conditions: a group who read a fotonovela with a Q&A section; a group who read a fotonovela without a Q&A section; a group who read an existing brochure used in the field, and a no message control group. The inclusion of this control group allowed for a posttest-only control group design, where a comparison with a pretest is not necessary [38].

#### 2.1.1. Materials

All materials were presented in Afrikaans, the dominant language of the target audience in the Western Cape Province. The fotonovela *Spyt kom te laat* [11,32] was used for both fotonovela message conditions. As discussed, this fotonovela did not have a Q&A section, so that had in one of the conditions in the current study. The factual information for the Q&A section was obtained from the South African National Council on Alcoholism and Drug Dependence (SANCA). See Figure 1 for the cover page and an example page of the fotonovela.

An existing brochure with similar content, *Die wrede feite oor tik* (*The cruel facts about MA*), was sourced from a non-governmental organization whose major objectives are the prevention and treatment of drug dependence. A self-administered questionnaire, also in Afrikaans and partly matching the questionnaire in the Davis and Jansen study [11], was used to collect the data. See Figure 2 for an example page of the existing brochure.

#### 2.1.2. Participants

Participants (*n* = 372) were from previously disadvantaged communities, mainly in the Western Cape towns of Ceres, Swellendam, and Malmesbury. The town of Calvinia, though situated in the Northern Cape Province, was also included as it borders the Western Cape Province. In Ceres, data were collected at a farm (Die Eike). Recruitment and data collection took place at participants’ place of work in Swellendam (comprising municipal workers). Learners and out-of-school youths from a childcare centre made up the participants from Malmesbury, with data gathered at a community centre. Any community member could attend an open town hall gathering in Calvinia where recruitment and data collection took place. Data collection also occurred at various schools in Calvinia (comprising learners). Community workers and healthcare providers in these areas helped with the recruitment and assisted with data collection. Participants were offered refreshments at some sites, while at other sites, lucky draws were held to show appreciation for participation. No other incentives were offered. Table 1 presents information on the participants’ gender, place of residence, level of education, and age group.

As Table 1 shows, 57.8% of the participants who filled in their gender were female (*n* = 215) and 40.4% were male (*n* = 141), with 16 missing values. Most of the participants resided in Calvinia (70.7%; *n* = 263). The level of education of participants was mostly low (80.7%; *n* = 267), many of the participants being learners and still at school (53.2%; *n* = 176). Participants who had a high level of education made up 19.3% (*n* = 64) of the cohort. In total, 56.9% of the participants were 19 years old or younger (*n* = 212), with the 35 years and older age group having the fewest participants (13.17%; *n* = 49).

#### 2.1.3. Measures

##### Health Literacy

To measure health literacy level, three items from Chew and colleagues [39] were presented to the participants: Health literacy 1 (“How often do you have someone help you read hospital materials?”), Health literacy 2 (“How often do you have problems learning about your medical condition because of difficulty understanding written information?”), and Health literacy 3 (“How confident are you filling out medical forms by yourself?”). Health literacy 1 and Health literacy 2 were measured using this 5-point scale (1 = *Never*; 2 = *Occasionally*, 3 = *Sometimes*, 4 = *Often*, 5 = *Always*). Health literacy 3 was measured using this 5-point scale: (1 = *Extremely*; 2 = *Quite a bit*, 3 = *Somewhat*, 4 = *A little bit*, 5 = *Not at all*). All scores were recoded such that a higher score indicated a higher level of health literacy. Reliability analysis showed that it was not possible to combine the reactions to these three items into one reliable scale: Cronbach’s alpha = 0.25. Leaving out one of the items did not lead to an acceptable Cronbach’s alpha either; the maximum attainable value was 0.52.

In a large scale validation study in a veterans population in the US, performed by Chew and colleagues [40] (*n* = 1.796), it was found that one of the three items developed by Chew and colleagues [39] was most effective in detecting inadequate health literacy: Health literacy 3 (“How confident are you filling out medical forms by yourself?”). The optimum screening threshold for this item proved to be the response of ‘somewhat’ or less. Based on the outcomes of other, more demanding and time consuming instruments for measuring health literacy level (S-TOFHLA and REALM), it was found that the proportion of participants who were correctly identified (true positives and true negatives) was 80%. No combination of the three items developed by Chew et al. significantly performed better than this single item (p. 564).

Considering these findings, in our study, the responses to Health literacy 3 (“How confident are you filling out medical forms by yourself?”) were used to create two groups: participants with a low level of health literacy (response to Health literacy 3: *Somewhat*, *A little bit*, or *Not at all*) and participants with a high level of health literacy (response to Health literacy *3*: *Quite a bit*’ or *Extremely*. Table 2 shows the distribution of participants with a low or a high level of health literacy in the four experimental conditions.

Relationship between level of health literacy and condition: not significant (χ^2^ (3) = 4.02, *p* = 0.26).

##### Knowledge Level

Knowledge level related to MA was measured with 10 statements which could be either true or false. The statements were chosen to create a relatively even representation of the short- and long-term physical and psychological effects related to MA use. Factual information about MA was sourced from SANCA. Each answer was scored as 1 = correct or 0 = incorrect (no answer was also regarded as incorrect). True statements were: “2. MA makes your body deteriorate fast”, “3. MA makes you aggressive”, “5. MA has a negative effect on your brain”, “8. MA causes you to imagine different things”, and “9. MA gives you false confidence”. False statements were “1. MA decreases your energy levels” (the correct answer is MA increases your energy levels), “4. MA causes you to think people are friendly and kind towards you” (the correct answer is MA causes you to think people are antagonistic towards you)”, “6. MA decreases your sex drive” (the correct answer is MA causes an increase in your sex drive), “7. MA makes your brain produce less dopamine” (the correct answer it makes your brain produce more dopamine), and “10. MA makes you gain weight” (the correct answer is it makes you gain weight). The total score for correct answers could vary between 0 and 10.

##### Attitudes and Intentions

The dependent variables we used included attitudes and intentions related to the socially responsible behaviour of not using MA, as well as speaking to a friend or family member who is involved with MA about their drug habit. As Davis and Jansen [11] argued, referring to Lubinga and colleagues [41], among others, the efficacy of mass media health communication campaigns can be enhanced by conversations about the core messages of these campaigns. Such conversations may lead to changes in relevant beliefs, attitudes, social norms, and behavioural intentions [42,43,44,45], thus fostering behaviour change. From a meta-analysis into the effects of campaign-generated conversations, Jeong and Bae [46] concluded that such conversations indeed have a positive, albeit small, effect on inducing campaign-targeted outcomes (p. 14).

Attitude toward MA Use

Attitude toward MA use was measured with three items. Attitude 1 was modelled on an item from the Risk Behavior Diagnosis (RBD) scale [47,48]: “To never use MA, is something I [..]”. A 5-point scale was used (1 = *do not feel strongly about at all* and 5 = *definitely feel strongly about*). The items for measuring Attitude 2 and Attitude 3 were modelled on items used to measure attitudes in the Theory of Planned Behaviour [49]: “For me to use MA will be [..]”, using two different 5-point scales (1 = *extremely unpleasant* and 5 = *extremely pleasant*) and (1 = *extremely bad* and 5 = *extremely good*). After reversing the scores for *Attitude 1*, reliability analysis showed that it was not possible to combine the reactions to these three items into one scale for attitude toward MA use: Cronbach’s alpha = 0.28. Leaving out one of the items did not lead to an acceptable Cronbach’s alpha; the maximum attainable value was 0.55.

Intention to Use MA

Intention to use MA was measured with three items. Intention 1 was modelled on an item from the RBD scale [47,48]: “I plan to not use MA in the future [..]”. A 5-point scale was used (1 = *strongly disagree* and 5 = *strongly agree*). Intention 2 was modelled on an item to measure behavioural intention from McMillian and Conner [50], (p. 1668): “Please indicate how often you plan to use MA over the next 6 months (if ever) [..]”. A 5-point scale was used ranging from 1 (*Never*) through 3 (*Every few months*) to 5 (*Every day*). Intention 3 was modelled on an item used to measure intention in conformity with the Theory of Planned Behaviour [49]: I plan to use MA in the future [..]”. A 5-point scale was used (1 = *extremely unlikely* and 5 = *extremely likely*). After reversing the scores for Intention 1, reliability analysis showed that it was not possible to combine the reactions to these three items into one scale for Intention to use MA: Cronbach’s alpha = 0.24. Leaving out one of the items did not lead to an acceptable Cronbach’s alpha; the maximum attainable value was 0.51.

Attitude toward Starting Conversations about MA

Attitude toward starting conversations about MA was measured with three items. The researcher-designed item Attitude 1 was measured using a 5-point scale (1 = *do not feel strongly about at all* and 5 = *definitely feel strongly about*): “To speak to a family member or friend who is involved with MA about their drug habit, is something I [..]. Attitude 2 and Attitude 3 were modelled on items used to measure attitudes in the Theory of Planned Behaviour [49]. Attitude 2 and Attitude 3 were measured using two different 5-point scales (1 = *extremely unpleasant* and 5 = *extremely pleasant*) and (1 = *extremely bad* and 5 = *extremely good*): “For me to talk to a family member or friend about MA will be [..]” *extremely unlikely* and 5 = *extremely likely*). Reliability analysis showed that the reactions to these three items could be combined into one scale for attitude toward starting conversations about MA: Cronbach’s alpha = 0.80.

Intention to Start Conversations about MA

Intention to start conversations about MA was measured with three items. The researcher-designed item Intention 1 was measured using a 5-point scale (1 = *strongly disagree* and 5 = *strongly agree*): “I plan to soon speak to a family member or friend who is involved with MA about their drug habit [..]”. Intention 2 and Intention 3 were modelled on items used to measure attitudes in the Theory of Planned Behaviour [49]. Intention 2 was measured using a 5-point scale (1 = *I definitely will not* and 5 = *I definitely will*): “I will make an effort to talk to a family member or friend about MA [..]”. Intention 3 was measured using a 5-point scale (1 = *extremely unlikely* and 5 = *extremely likely*): “I plan to talk to a family member or friend about MA [..]”. Reliability analysis showed that the reactions to these three items could be combined into one scale for intention to start conversations about MA: Cronbach’s alpha = 0.80.

##### Health Message Preference

For measuring health message preference, one question was used: “Do you prefer to read a message about MA in the form of a booklet such as *Spyt kom te laat*, or rather a brochure like *Die wrede feite oor tik*?

#### 2.1.4. Procedure

At the various data collection sites, the researcher explained to participants what the study was about, after which written informed consent was obtained. Participants were then randomly divided into four groups: (1) fotonovela with a Q&A section (*n* = 99); (2) fotonovela without a Q&A section (*n* = 82); (3) existing brochure (*n* = 94); and (4) control (*n* = 97). Each participant was given an envelope containing either: (1) the fotonovela without a Q&A section, the existing brochure, and an accompanying questionnaire, or (2) the fotonovela with a Q&A section, the existing brochure, and an accompanying questionnaire. All participants answered questions about health literacy, knowledge level, attitudes and intentions, and health message preference.

All participants who received either one of the fotonovelas or the existing brochure (conditions 1–3) were asked to take their time when reading the health documents. Group allocation determined which health document participants had to read prior to completing the questionnaire. They were instructed to put the health document they had to read back in the envelope before completing the questionnaire. After this, they were asked to look at the health message in the other format and to answer the question about which format they preferred. The group who first read the fotonovela with a Q&A section had to choose between this document and the existing brochure, the group who first read the fotonovela without a Q&A section had to choose between this document and the existing brochure, and the group who first read the existing brochure had to choose between this document and the fotonovela with a Q&A section. The no message control group (condition 4), after first completing the questionnaire, then looked at the existing brochure and the fotonovela without a Q&A section and asked for their preference.

#### 2.1.5. Statistical Analyses

Statistical procedures for this part of the study included conducting univariate analyses (ANOVAs), a *t*-test, and chi-square analyses [51]. ANOVAs were deemed appropriate to use here as we wanted to find the effects of several independent variables (condition, gender, age group, and level of health literacy (low/high)) on different dependent variables (knowledge level, attitude toward starting conversations about MA and intention to start conversations about MA). Given the impossibility to create dependent variables related to attitude toward MA usage and intention toward MA usage with an acceptable level of reliability (see above), these variables were excluded from further analyses. A t-test was conducted to find possible differences in knowledge level between two conditions (fotonovela with and without Q&A section). To identify any existing relationships between health message preference on the one hand and condition, gender, age group, and level of health literacy (low/high) on the other hand, chi-square analyses were conducted (all these variables were categorical). SPSS Statistics 27 (IBM, Armonk, NY, USA) was used for all statistical analyses.

### 2.2. Method for the Part of the Study Related to RQ2

An experimental design was followed including four message conditions (a group who first assessed the fotonovela with Q&A section, then looked at the existing brochure; a group who first assessed the fotonovela without Q&A section, then looked at the existing brochure; a group who first assessed the existing brochure, then looked at the fotonovela with Q&A section; and a group who just looked at the fotonovela with Q&A section and the existing brochure).

#### 2.2.1. Materials

All materials were presented in Afrikaans to the healthcare providers. The same materials used for the part of the study related to RQ1 were used in this part of the study (see above).

#### 2.2.2. Participants

Participants (*n* = 75) were from the towns of Ceres (*n* = 49) and Prince Alfred Hamlet (*n* = 26). In both towns, data were collected among groups of local healthcare providers (nurses and community health workers) during regional meetings. The local Department of Health in these areas helped with the recruitment and assisted with data collection. Participants were given refreshments. No incentives were offered. Table 3 presents information on the participants’ gender, place of residence, level of education, and age group.

As Table 3 shows, almost all participants were female (94.7%; *n* = 71). As many of the participants were professional nurses, the level of education of participants was mostly high (60%; *n* = 45), although there were participants who had a low level of education as well (36%; *n* = 27). Community health workers, who formed part of the cohort, often do not have a formal education and receive in-service training. Most participants were 35 years and older (65.3%; *n* = 49).

#### 2.2.3. Measures

##### Health Message Preference

For measuring health message preference, the same question was posed to participants as used for the part of the study related to RQ1 (see above).

##### Evaluation of Fotonovela Use

For measuring the evaluation of possible fotonovela use in a healthcare setting, three researcher-designed items were included: “In my role as healthcare worker or facility manager, I will recommend this photo novel for my clients who want to learn more about MA”, “In my role as healthcare worker or facility manager, I find this photo novel applicable to convince clients of the dangers of MA”, and ““In my role as healthcare worker or facility manager, I find this photo novel applicable to give clients the correct information about MA”. For each of these items, a 5-point scale was used (1 = *strongly disagree* and 5 = *strongly agree*). Reliability analysis showed that the reactions to these three items could be combined into one scale for evaluation of possible fotonovela use: Cronbach’s alpha = 0.85.

#### 2.2.4. Procedure

A similar procedure was followed as in the part of the study related to RQ1, in terms of explaining the context of the study, obtaining informed consent, and randomly dividing the participants into four groups. The numbers of participants per condition were as follows: (1) first assessing the fotonovela with Q&A section, then looking at the existing brochure (*n* = 20); (2) first assessing the fotonovela without Q&A section, then looking at the existing brochure (*n* = 20); (3) first assessing the existing brochure, then looking at the fotonovela with Q&A section (*n* = 20); and (4) just looking at the fotonovela with Q&A section and the existing brochure (*n* = 15). After this, participants were asked to complete the questions about message preference and possible use of fotonovelas in a health setting.

#### 2.2.5. Statistical Analyses

To identify a possible relationship between condition and health message preference, a chi-square test was performed (both variables were categorical). An ANOVA was conducted to test the effect of the independent variable condition on the dependent variable evaluation of fotonovela use. Again, SPSS Statistics 27 (IBM, Armonk, NY, USA) was used.

## 3. Results 

### 3.1. Results for the Part of the Study Related to RQ1

For RQ1, we wanted to know how a fotonovela (with or without a Q&A section) compared to an existing brochure in affecting MA knowledge, attitudes, intentions, and preference of readers from different age groups, with a low or a high level of health literacy.

#### 3.1.1. Knowledge Level

The overall mean score for knowledge level was 7.07 (*SD* = 1.31; all variables were measured on a 10-point scale). The ANOVA was conducted to test the effects of condition, gender, age group, and level of health literacy (low/high) on knowledge level revealed no significant main effect of condition: F(3,247) = 1.74; *p* = 0.16. There were also no significant main effects of gender (F(1,247) = 2.93; *p* = 0.09), age group (F(2,247) = 1.13; *p* = 0.32) or level of health literacy (F(1,247) = 0.02; *p* = 0.89), nor were there two-way interaction effects of condition and gender (F(3,247) = 0.71; *p* = 0.55), condition and age group (F(6,247) = 0.93; *p* = 0.48), or condition and level of health literacy (F(3,247) = 1.21; *p* = 0.30). There was one significant three-way interaction effect, of condition, age group, and level of health literacy (F(6,247) = 2.47; *p* = 0.02; η^2^ = 0.06).

Follow up analyses revealed that in the age group ‘35 or older’, there was a significant interaction effect of condition and level of health literacy (F(3,28) = 3.61; *p* = 0.02; η^2^ = 0.28). In the ‘fotonovela with Q&A section’ condition, the ‘low level of health literacy’ group (*M* = 7.67; *SD* = 2.31) outperformed the ‘high level of health literacy’ group (*M* = 7.00; *SD* = 1.63). In the ‘fotonovela without Q&A section’ condition, the ‘low level of health literacy’ group (*M* = 9.00; *SD* = 0.00) also outperformed the ‘high level of health literacy’ group (*M* = 6.75; *SD* = 1.50). In the ‘existing brochure’ condition, however, the ‘high level of health literacy’ group (*M* = 8.73; *SD* = 1.19) outperformed the ‘low level of health literacy’ group (*M* = 5.50; *SD* = 0.71). Additionally, in the control condition, the ‘high level of health literacy’ group (*M* = 6.75; *SD* = 0.96) outperformed the ‘low level of health literacy’ group (*M* = 6.25; *SD* = 0.96).

In the age group ‘19 or younger’, no significant interaction effects of condition and level of health literacy were found (F(3,167) = 0.99; *p* = 0.40); nor were there significant interaction effects in the age group ‘20–34’ of condition and level of health literacy (F(3,73) = 0.84; *p* = 0.48).

The *t*-test conducted to find possible differences in knowledge level between the fotonovela versions with a Q&A section (*M* = 7.38; *SD* = 1.37) and without a Q&A section (*M* = 7.20; *SD* = 1.40) revealed no significant difference: t(186) = 0.85; *p* = 0.84.

#### 3.1.2. Attitude and Intention toward MA Usage

Since it proved to be impossible to create dependent variables with an acceptable level of reliability (see above), no results can be reported here.

#### 3.1.3. Attitude toward Starting MA Conversations

The overall mean score for attitude toward starting MA conversations was 3.61 (*SD* = 1.28; all variables were measured on a 5-point scale). There was a main effect of condition on attitude toward starting MA conversations that just reached statistical significance (F(3,301) = 2.69; *p* = 0.046). Post hoc tests (Bonferroni) revealed one significant difference between the four conditions: participants who had read the fotonovela with Q&A section had a significantly more positive attitude toward starting MA conversations (*M* = 3.94; *SD* = 1.05) than participants in the control condition (*M* = 3.44; *SD* = 1.30) (*p* = 0.03). The main effect of condition on attitude toward starting MA conversations was qualified by a two-way interaction effect that bordered on statistical significance, of condition and level of health literacy F(3,301) = 2.63; *p* = 0.05). Follow up analyses revealed that in the group of participants with a low level of health literacy there was no significant effect of condition. In the group of participants with a high level of health literacy, however, such a significant effect was found (F(3,205) = 4.29; *p* < 0.01; η^2^= 0.06). Post hoc tests (Bonferroni) revealed two significant differences between the four conditions: participants who had read the fotonovela with Q&A section had a significantly more positive attitude toward MA conversations (*M* = 4.10; *SD* = 0.92) than participants in the control condition (*M* = 3.41; *SD* = 1.46) (*p* = 0.03) and than participants who had read the fotonovela without Q&A section (*M* = 3.35; *SD* = 1.46) (*p* = 0.02). There were no significant three-way interaction effects.

#### 3.1.4. Intention to Start Conversations about MA

The overall mean score for intention to start conversations about MA was 3.80 (*SD* = 1.20; all variables were measured on a 5-point scale). There was no significant main effect of condition on intention to start conversations about MA (F(3,306) = 0.87; *p* = 0.45), nor were there significant two-way or three-way interaction effects.

#### 3.1.5. Health Message Preference

Overall, a preference was found for the fotonovela: preference for fotonovela with or without a Q&A section: *n* = 219; preference for existing brochure: *n* = 79; no preference: *n* = 38; missing values: *n* = 36. The preference for the fotonovela over the existing brochure was significant: χ^2^ (1) = 65.77 (*p* < 0.001). Chi-squares were calculated to find possible associations between health message preference on the one hand, and condition, gender, age group, and level of health literacy (low/high) on the other hand. No such significant relationships were found: health message preference and condition: χ^2^ (6) = 9.80, *p* = 0.13; health message preference and gender: χ^2^ (2) = 2.71, *p* = 0.26; health message preference and age group: χ^2^ (4) = 4.23, *p* = 0.38; health message preference and level of health literacy: χ^2^ (2) = 4.3, *p* = 0.80.

### 3.2. Results for the Part of the Study Related to RQ2

For RQ2, we wanted to find out how a MA fotonovela compared to an existing health brochure and how it would be evaluated when assessed by healthcare providers.

#### 3.2.1. Health Message Preference

Overall, a preference was found for the fotonovela: preference for the fotonovela with or without a Q&A section: *n* = 61; preference for the existing brochure: *n* = 7; no preference: *n* = 2; missing values: *n* = 5. The preference for the fotonovela over the existing brochure was significant: χ^2^ (1) = 42.88 (*p* < 0.001). A chi-square test was calculated to find a possible association between health message preference, on the one hand, and condition on the other hand. No significant relationship between health message preference and condition was found: χ^2^ (6) = 9.43, *p* = 0.15.

#### 3.2.2. Evaluation of Fotonovela Use 

The overall mean score for evaluation of fotonovela use in a healthcare setting was 4.73 (*SD* = 0.39; missing values: *n* = 1). The ANOVA that was conducted to test the effect of condition revealed no significant effect: F(3,69) = 0.85; *p* = 0.47.

## 4. General Discussion

This study aimed to collect more empirical data to reach an informed conclusion about the effects of a fotonovela intended to convincingly convey health information about MA in South Africa. The first part of the study focused on farm workers, municipal workers, and community members from previously disadvantaged communities. For all three age groups and for both levels of health literacy distinguished, we found a clear preference for the fotonovela format over an existing brochure with a comparable content. Furthermore, we found that in the group of older participants, the fotonovela led to significantly higher knowledge scores for readers with a low level of health literacy than for readers with a high level of health literacy; in the other conditions, it was the opposite. Contrary to expectations based on previous research [35], the inclusion of a Q&A section did not lead to a significantly higher knowledge score. Perhaps readers found the information in the Q&A section, which was more clinical in nature compared to the photo story itself, relatively difficult to follow. However, readers of the fotonovela with Q&A section had the most positive attitude toward starting a conversation about MA. This was especially the case for readers with a high level of health literacy.

The second part of the study targeted healthcare providers, a group that was not included in the Davis and Jansen study [11]. This group also clearly proved to prefer the fotonovela format to the existing brochure. In addition, their evaluation of the potential use of fotonovelas in a health care setting proved very positive.

The findings from the current study partly do and partly do not confirm those from the study by Davis and Jansen [11]. Again, a clear preference was found for reading health information about MA in fotonovela format rather than in traditional brochure format. This is an important outcome. According to McGuire’s classic input–output framework for constructing persuasive messages [52], (p. 134), a series of thirteen steps is necessary for any message to be persuasive. Immediately after the first two steps, ‘tuning in’ and ‘attending to the communication’, and before the other steps including ‘comprehending its contents’, ‘agreeing with the communication’s position, ‘decision to act on the basis of the retrieved information’, and ‘acting on it’ comes ‘liking it, maintaining interest in it’. If large portions of the target audience do not want to read the health message intended to serve their interests, or if they decide to stop reading prematurely, then all other qualities of the form and content of the message no longer matter.

Additionally, in both studies, reading the fotonovela had a positive effect on the level of knowledge in some situations. In the earlier study, readers of the fotonovela performed better than participants in the control condition, but not better than readers of the traditional brochure with which it was compared. In the current study, the fotonovela led to a higher knowledge score than the existing brochure, but only for older participants with a low level of health literacy.

### 4.1. Strengths and Limitations

The first and most important strength of this study is that it helped to reach an informed and positive conclusion about the effects of a fotonovela designed to convey health information about MA. Both the earlier study by Davis and Jansen [11] and this new study found that the vast majority of the target audience prefers to read about MA in fotonovela form. Moreover, the current study shows that health professionals have high confidence in the ability of fotonovelas to disseminate relevant health information.

Second, the distinction we were able to make between various groups of participants with different levels of health literacy allowed us to discover that especially older people with low levels of health literacy can benefit from a fotonovela as a means to increase their knowledge about an important health topic. Given that older adults in particular are most likely to experience the consequences of low health literacy [34], we consider this an important finding.

This study also has some limitations. Low correlations of items intended to measure outcomes attitude toward MA use and intention toward MA use made it impossible to create these dependent variables. Furthermore, only one fotonovela, albeit with and without a Q&A section, about only one topic was compared to only one example of an existing brochure about that same topic. This necessarily limits the generalizability of the results. Finally, there were only a limited number of participants aged 35 or older. Conceivably, participation by more people of a considerably higher age would have led to even clearer effects of health literacy level.

### 4.2. Implications for Practice

Given that recent statistics show that many adults in the Western Cape Province have low health literacy levels, and in light of current MA abuse campaigns often focussing on reaching older audiences in the province, fotonovelas could be useful to create messages specifically geared towards increasing the knowledge levels about MA of this cohort. Fotonovelas could thus be used to promote and boost existing, more established MA abuse prevention efforts or campaigns currently being rolled out in the Western Cape Province. Actually, the Department of Health has already distributed the fotonovela *Spyt kom te laat* [32] in three different versions, each intended for native speakers of the most spoken languages in the province: Afrikaans, English, and isiXhosa.

Another new approach could be to present short photo stories containing MA information online—for reading on a computer but also on a smartphone. For the latter, photo stories could be disseminated in the form of an interactive, digital booklet using existing messaging apps such as the commonly used WhatsApp platform.

### 4.3. Implications for Research

In this study, specifically older people with low levels of health literacy benefitted from reading about MA in fotonovela format. It is not really clear how this finding may be explained. In future studies, attention could therefore perhaps shift to a more theoretical perspective to try and find an answer, for instance, focusing on the role of identification, transportation, and perceived similarity in overcoming resistance to persuasive messages contained in a narrative [53,54]. Recent empirical research suggests that, for instance, perceived age similarity of the reader to that of the protagonist can play an important role in how a message is received, with younger and older participants differing on a number of outcomes [55].

## 5. Conclusions

The findings of the current study, added to the results of Davis and Jansen [11], argue for considering fotonovelas as a serious option to effectively communicate convincing health information about MA use to target audiences in the Western Cape Province. Taken together, these two studies provide further support for the position that using narratives in health communication is a promising strategy for improving both its appeal and its effectiveness. Of course, the problem of MA abuse is multi-layered and complex. Much more is needed to make health communication about MA successful than distributing a fotonovela designed to convince people to stop using MA, to seek out treatment options, or to not use this drug in the first place. Still, circulating a fotonovela—or using it as an educational tool in a healthcare setting—could provide the spark to reignite prevention efforts in the province and the country.

## Figures and Tables

**Figure 1 ijerph-18-06334-f001:**
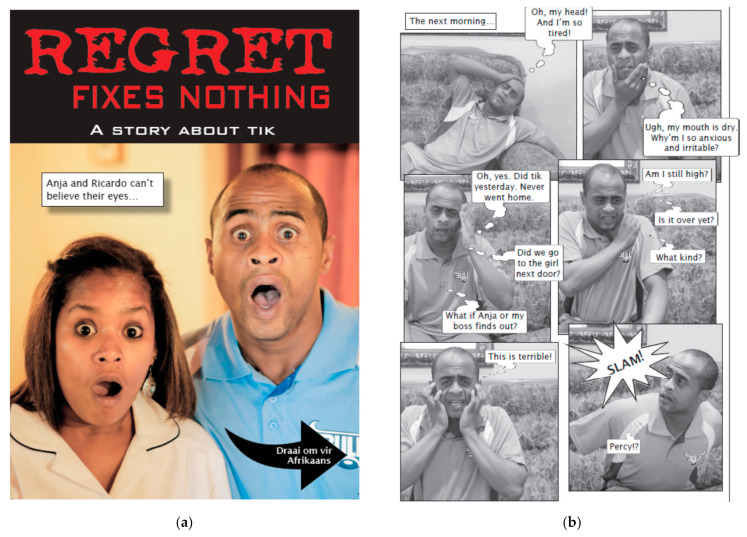
Cover page (panel (**a**)) and an example page (panel (**b**)) of the fotonovela version in English (in this study, only the Afrikaans version was used). As mentioned earlier, the street name for MA in South Africa is ‘tik’.

**Figure 2 ijerph-18-06334-f002:**
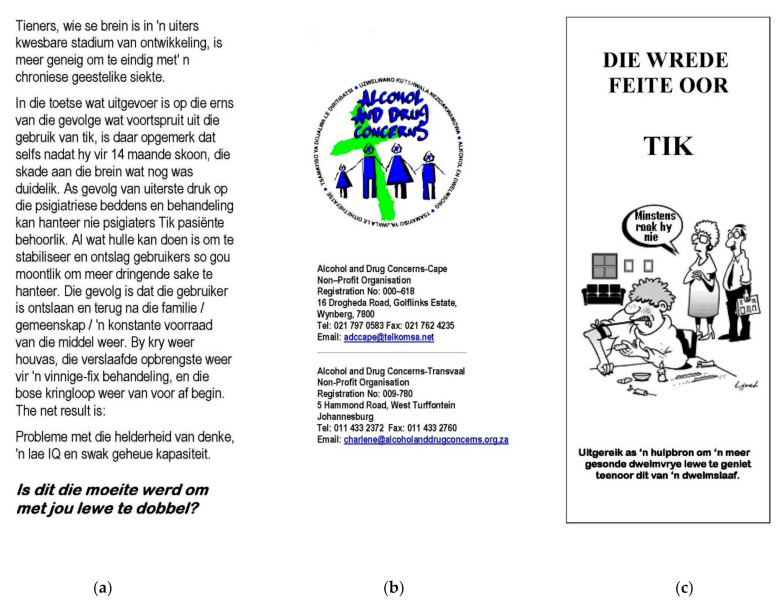
Example page of the existing brochure used in this study (Afrikaans version; there was no English version). Summary of the content in panel (**a**): This section discusses the dangers of MA use. Summary of the content in panel (**b**): This section reflects the logo and contact details of the non-governmental organization who developed the brochure. Summary of the content in panel (**c**): This section shows the cover page. Translation of the text above the cartoon/image: ‘The cruel facts about MA’.

**Table 1 ijerph-18-06334-t001:** Descriptive statistics for the part of the study related to RQ1.

	Fotonovela with Q&A Section	Fotonovela without Q&A Section	Existing Brochure	Control Condition	Total
	(*n* = 99)	(*n* = 82)	(*n* = 94)	(*n* = 97)	(*n* = 372)
Gender					
Male	*n* = 31	*n* = 38	*n* = 30	*n* = 42	*n* = 141
Female	*n* = 66	*n* = 40	*n* = 60	*n* = 49	*n* = 215
Missing values	(*n* = 2)	(*n* = 4)	(*n* = 4)	(*n* = 6)	(*n* = 16)
Place of residence					
Ceres	*n* = 7	*n* = 0	*n* = 2	*n* = 0	*n* = 9
Swellendam	*n* = 16	*n* = 9	*n* = 13	*n* = 22	*n* = 60
Malmesbury	*n* = 8	*n* = 11	*n* = 13	*n* = 8	*n* = 40
Calvinia	*n* = 68	*n* = 62	*n* = 56	*n* = 67	*n* = 263
Level of education					
Low (still at school; left school before Grade 10, 11, or 12)	*n* = 78	*n* = 59	*n* = 60	*n* = 70	*n* = 267
High (high school diploma or degree after high school)	*n* = 10	*n* = 13	*n* = 24	*n* = 17	*n* = 64
Missing values	(*n* = 11)	(*n* = 10)	(*n* = 10)	(*n* = 10)	(*n* = 41)
Age Group					
19 or younger	*n* = 51	*n = 53*	*n = 50*	*n* = 58	*n* = 212
20–34 years	*n* = 29	*n* = 19	*n* = 22	*n* = 26	*n* = 96
35 years or older	*n* = 17	*n* = 6	*n* = 17	*n* = 9	*n* = 49
Missing values	(*n* = 2)	(*n* = 4)	(*n* = 5)	(*n* = 4)	(*n* = 15)

**Table 2 ijerph-18-06334-t002:** Health literacy levels of participants for the part of the study related to RQ1.

	Fotonovela with Q&A Section	Fotonovela without Q&A Section	Existing Brochure	Control Condition	Total
	(*n* = 99)	(*n* = 82)	(*n* = 94)	(*n* = 97)	(*n* = 372)
Level of health literacy					
Low	*n* = 40	*n* = 38	*n* = 33	*n* = 48	*n* = 159
High	*n* = 58	*n* = 43	*n* = 58	*n* = 49	*n* = 208
Missing values	(*n* = 1)	(*n* = 1)	(*n* = 3)	(*n* = 0)	(*n* = 5)

**Table 3 ijerph-18-06334-t003:** Descriptive statistics for the part of the study related to RQ2.

	Total
	(*n* = 75)
Gender	
Male	*n* = 3
Female	*n* = 71
Missing values	(*n* = 1)
Place of residence	
Ceres	*n* = 49
Prince Alfred Hamlet	*n* = 26
Level of education	
Low (still at school; left school before Grade 10, 11, or 12)	*n* = 27
High (high school diploma or degree after high school)	*n* = 45
Missing values	(*n* = 3)
Age Groups	
19 or younger	*n* = 0
20–34 years	*n* = 25
35 years and older	*n* = 49
Missing values	(*n* = 1)

## Data Availability

Data supporting the reported results can be found at https://www.careljansen.nl/18-5-2021_BD_CJ_Tik_Fotonovela_Experiment.sav (accessed on 15 May 2021).

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
