# Peer review of "Deploying a Fotonovela to Combat Methamphetamine Abuse among South Africans with Varying Levels of Health Literacy"

_ijerph, 2021, doi:10.3390/ijerph18126334_

Round 1

Reviewer 1 Report

In this manuscript, the authors reported data regarding the use of fotonovela to contrast the use of methamphetamine abuse in South Africa. I think that this paper is very interesting and also the methodologies are appropriate. Moreover it could have several implications on African population with low levels of literacy. I really appreciate that the authors are aware on the limitations of their study discussing them in the text. II have a only few suggestions to improve the manuscript:

  • Did you find any correlation stratifying data according to the gender?
  • Is there any correlation with the degree of literacy?

Reviewer 2 Report

The manuscript is well structured and easy to read.

  There were no major shortcomings that would keep it from being considered for publication.  The research problem and research questions are clear and addresses the major facets of the fotonovela study. 

The methodology is well presented, and the different scales used are appropriate and described adequately. 

The findings and discussion are informative. Overall, the manuscript is strong and well-written.  This reviewer has some recommendations to strengthen the article further and to bring more clarity to it. 

The authors should address contributing factors, other than health literacy, that contribute to the “hampering MA prevention efforts.  Identifying them should suffice.   Also, there should be more discussion of the origins and use of fotonovelas in health promotion.

  Not all readers may be familiar with this background.  In this discussion, how the fotonovela has been used for drug use prevention and intervention elsewhere should be included.

  Also, it is unclear whether “knowledge level related to MA” using the true and false 10 statements were only measured at the beginning of the study as well as at the end.

 This should be clarified. 

The manuscript should be published with the minor revisions suggested.

Reviewer 3 Report

Counteracting drug abuse is very important and topical from the point of view of health and the proper functioning of society. Research indicates that this problem occurs in both rich and developing countries. However, counteracting this phenomenon may be more or less effective, depending on the different social and economic characteristics of the countries and areas concerned. Therefore, the issues contained in the manuscript may be interesting for potential readers, and I consider them essential for science. The manuscript structure and literature selection are correct. However, I have some comments for the authors:

  1. In the "Introduction" section, I suggest adding point 1.7. The authors will briefly present the further structure of the manuscript.
  2. The authors applied selected statistical tests to data analysis. However, there is no short description of the statistical methodology used, together with an indication of the program used for the calculations.
  3. I suggest extending the section "Conclusions."

Congratulations to the authors of the idea and good luck in further research!
